# Equibiaxial Strained Oxygen Adsorption on Pristine Graphene, Nitrogen/Boron Doped Graphene, and Defected Graphene

**DOI:** 10.3390/ma13214945

**Published:** 2020-11-04

**Authors:** Li-Hua Qu, Xiao-Long Fu, Chong-Gui Zhong, Peng-Xia Zhou, Jian-Min Zhang

**Affiliations:** 1School of Science, Nantong University, Nantong 226019, China; chgzhong@ntu.edu.cn (C.-G.Z.); ntzhoupx@ntu.edu.cn (P.-X.Z.); 2Xi’an Modern Chemistry Research Institute, Xi’an 710065, China; fuxiaolong204@163.com; 3College of Physics and Information Technology, Shaanxi Normal University, Xi’an 710062, China; jmzhang@snnu.edu.cn

**Keywords:** oxygen adsorption, graphene, equibiaxial strain

## Abstract

We report first-principles calculations on the structural, mechanical, and electronic properties of O_2_ molecule adsorption on different graphenes (including pristine graphene (G–O_2_), N(nitrogen)/B(boron)-doped graphene (G–N/B–O_2_), and defective graphene (G–D–O_2_)) under equibiaxial strain. Our calculation results reveal that G–D–O_2_ possesses the highest binding energy, indicating that it owns the highest stability. Moreover, the stabilities of the four structures are enhanced enormously by the compressive strain larger than 2%. In addition, the band gaps of G–O_2_ and G–D–O_2_ exhibit direct and indirect transitions. Our work aims to control the graphene-based structure and electronic properties via strain engineering, which will provide implications for the application of new elastic semiconductor devices.

## 1. Introduction

Due to good mechanical and electrical characteristics, graphene-based materials have attracted significant attention [1,2,3,4,5,6]. For instance, graphene possesses ultra-large surface area (about 2630 m^2^ g^−1^), high mechanical stiffness, good electrical conductivity, low Johnson noise, and excellent thermal and optical properties. In order to make the most of the potential values of graphene sensors, it is significant to understand the coactions between adsorbed molecules and graphene. It has been identified that gas molecules can effectively change the electronic and magnetic properties of graphene and graphene nanoribbons [7,8]. Furthermore, molecular oxygen has multifarious chemical interactions with aromatic crystals [9,10] and carbon nanotubes [11], and O_2_ adsorption is a rather extensive, effective means for modifying these species [9,11]. In general, deformations of aromatic systems produce more sharp interactions with O_2_ [10]. The current research reveals that the types of graphene obviously affect the systems adsorbed by O_2_. It is known that the graphenes probably have defects resulted from multiple fabrication methods, and graphene doping is everywhere. Currently, research on defects and doping of graphene sensors has attracted much attention because of its numbing applications. On the other hand, strain exists objectively in constructed graphene structures, showing up in the ridges and buckling generation [12,13]. The maximum strain applied in graphene is 30% [14], showing large stiffness. In this case, there are many investigations about strain effects on the electronic characters of graphene and graphene nanoribbons [12,13,15,16,17]. In addition, decoration/passivation and strain effects on graphene work function are also research hotspots. Since graphene oxide (GO) derives from the monolayer of graphite oxide, it is interesting to know how the O_2_ molecule will affect the characters of different graphenes. Actually, the characters of the deformed GO are very significant for the applications of two-dimensional elastic oxide devices.

In this work, we performed first-principles simulations to investigate the equibiaxial strain effects on the interactions between the O_2_ molecule and various graphenes (including pristine graphene, N(nitrogen)/B(boron)-doped graphene and defective graphene). We calculated the pristine graphene as a contrast. The graphenes are doped by N and B atoms, representing the most widely-applied n-type and p-type dopants. For the defective graphene, there is only one missing atom in each supercell in consideration of reducing complexity. The research aims to study the influence of equibiaxial strain effects on the structural, mechanical, and electronic properties of O_2_ molecules adsorption on various graphenes.

## 2. Computational Methods

The first-principles calculations were performed through applying the Vienna ab-initio simulation package (VASP) [18,19,20,21,22] within the projector augmented-wave (PAW) approach [23]. The electron exchange-correlation was treated by utilizing the Perdew–Burke–Ernzerhof (PBE) formulation of the generalized gradient approximation (GGA). An energy cutoff of 450 eV for the plane-wave basis set was applied for all calculations. The single layer graphene was modeled with a unit cell containing 32 carbon atoms, which was separated with a 20 Å vacuum layer in the *z*-axis direction. The N/B-doped graphene was modeled by replacing one carbon atom with one N/B atom in the unit cell. The Monkhorst–Pack scheme [24] of the k-point mesh was adopted for integration in the Brillouin zone. With these computational parameters, the total ground state energy was converged within 10^−4^ eV per formula unit. The equibiaxial strain (ε) is defined as [25]
(1)$$\varepsilon =(a-a_{0})/a_{0}=(b-b_{0})/b_{0}$$
where 
a, b
and
$a_{0},b_{0}{}_{}$ are the deformed are the deformed (stretched or shrunken) and initial equilibrium lattice constants of the supercell in x, y directions, and ε varies from −10% to 10% with a step of 1%.

## 3. Results and Discussion

### 3.1. Geometric Structures and Stability

N and B belong to the second group elements in the periodic table, which have 2s2p configurations for electrons. They are the nearest elements for C. The extension of electrons of N, C, and B is similar. Therefore, when N and B are doped in graphene without strain, there is no local distortion. O_2_ can be physically absorbed on a N/B-doped graphene, which is the same as O_2_ on B-doped graphite [26]. Moreover, the physical adsorption does not significantly change the structure of N/B-doped graphene. Meanwhile, it was found that the structures of O_2_ adsorption on different graphenes are comparatively different under the equibiaxial strain. The top and side views of O_2_ adsorption on different graphenes under equibiaxial strains of −5%, 0%, 5%, and 8% are shown in Figure 1: O_2_ adsorption on pristine graphene (G–O_2_) in Figure 1(a1–d2); O_2_ adsorption on N-doped graphene (G–N–O_2_) in Figure 1(e1–h2); O_2_ adsorption on B-doped graphene (G–B–O_2_) in Figure 1(i1–l2); and O_2_ adsorption on defected graphene (G–D–O_2_) in Figure 1(m1–p2). The green, red, yellow, and blue balls denote the C, O, N, and B atoms, respectively, and the insides of dashed lines denote the unit cell of each system. At the beginning, an O_2_ molecule is placed at the top center of the supercell. After relaxation, for the G–O_2_, the O_2_ is parallel to graphene plane with or without strain, and the O_2_ molecule prefers to be located in the middle of a C6 ring (see Figure 1(a1–d2)). For G–N–O_2_, the O_2_ molecule shifts a bit from the top center with the two O atoms at different planes under compressive strain (see Figure 1(e1,e2)), while the O_2_ molecule is almost parallel to the N-doped graphene plane under no strain or tensile strain (see Figure 1(f1–h2)). The configurations of G–B–O_2_ under strains are similar to those of G–N–O_2_ (see Figure 1(i1–l2))]. For G–D–O_2_, the two O atoms are at different planes with or without strain. Furthermore, we noted that when the tensile strain is increased to 8%, the O–O bond is almost vertical to the defected graphene sheet (see Figure 1(p1,p2)).

We then investigate the binding energies ($E_{b}$) to clarify the stability of the O_2_ molecule adsorption on different graphenes (see Figure 2a). The binding energy can be obtained from total energy calculations as follow:(2)$$E_{b}=(E_{O}{}_{{}_{2}}+E_{0}-E_{1})/n$$
where
${E_{O}}_{2}$
is the total energy of an oxygen molecule in vacuum; $E_{1}$ and $E_{0}$ are total energies of graphene with and without molecule O_2_ adsorbed on it, respectively; and *n* is the number of all atoms of each adsorption system. Therefore, a positive binding energy state shows that the interactions between the O_2_ molecule and the graphene sheet are attractive, indicating the exothermic adsorption process. In Figure 2a, at the same strain, the binding energy of the G–D–O_2_ is slightly higher than those of G–O_2_, G–N–O_2_, and G–B–O_2_ when the magnitude of the compressive strain is larger than 2%. Subsequently, all the curves tend to zero. Thus, the energy cost effect induced by the graphene defect is minimal, especially in the aspect of the defective graphene’s structural integrity. The large binding energy also indicates that the O_2_ molecule can be pinned chemically at the defect on graphene, which is shown in Figure 1(p2) as the quantized plastic flow observed in gold nanowires [27]. The shortest distance of the O atom to different graphenes as the function of equibiaxial strain is shown in Figure 2b. We noted that all the distances vary in a nonmonotonic way when the magnitude of the compressive strain is larger than 2%, and then they tend to be constant with the decreasing (increasing) compressive (tensile) strain. Above all, the stability of O_2_ adsorption on different graphenes can be enhanced greatly by the compressive strain larger than 2%. However, it will be reduced under compressive strain smaller than −2% and tensile strain. This is assigned to the changes of structure configuration. Under smaller compressive strain and a tensile strain, the out-of-plane fold can be reduced faintly and partially by the in-plane lattice constant expansion, which recovers the π electron characteristics of graphene, indicating the stability of the binding strength (see Figure 3).

### 3.2. Mechanical Properties

To describe the mechanical properties of G–O_2_, G–N–O_2_, G–B–O_2_, and G–D–O_2_, the stress and Young’s modulus are determined. Firstly, the stress–strain curves for the G–O_2_, G–N–O_2_, G–B–O_2_, and G–D–O_2_ are shown in Figure 3. It is apparent that the stress–strain relation is not very dependent on the structures of the graphene sheets. We can see that when the compressive strain decreases to −2%, it becomes a constant with the decreasing (increasing) compressive (tensile) strain. The nonlinear trend of the stress–strain curves under compressive strain corresponds to graphene wrinkling (see Figure 1(a2,e2,i2,m2)).

Young’s modulus is measured by the slope of the stress–strain curve. From Figure 4, the effects of the tube equibiaxial strain on Young’s modulus for G–O_2_, G–N–O_2_, G–B–O_2_, and G–D–O_2_ are obviously observed. For each configuration, Young’s modulus changes nonlinearly, firstly with the decreasing compressive strain and failure at −2%. It is a consequence of the nonlinear response of the lateral “springs” that prevents O_2_ adsorption configurations from distortion during compression or tension. In addition, we can see that Young’s modulus is also independent of the types of the graphenes.

### 3.3. Electronic Properties

Since the equilibrium bonding configuration is presumably determined by the distribution and hybridization of energy levels, we computed the corresponding spin-polarized band structures under equibiaxial strain, which is illustrated in Figure 5: G–O_2_ (Figure 5a–f), G–N–O_2_ (Figure 5g–l), G–B–O_2_ (Figure 5m–r), and G–D–O_2_ (Figure 5s–x). The figures from the left columns to right columns correspond to the band structures calculated at the representative strains of −8%, −5%, −2%, 0%, 6%, and 10% in turn. The red and blue lines represent spin-up and spin-down bands, respectively. The Fermi level was set to zero, and it is represented by the horizontal dashed line. Firstly, it was found that G–O_2_ (see Figure 5a–f) experiences a gap transition between direct and indirect. For the compressive strains larger than 5% (see Figure 5a,b), the G–O_2_ shows a direct semiconductor character. At a −5% strain (see Figure 5b), the spin-up state presents a direct band gap character, while the spin-down state becomes an indirect one. With the decreasing (increasing) compressive (tensile) strain (see Figure 5), both the spin-up and spin-down states reveal a direct character again. For the G–N–O_2_ (see Figure 5g–l), some bands appear in the band gap region, and a few bands pass through the Fermi level, namely that the G–N–O_2_ presents a metal state, and its electronic properties show less dependence on the equibiaxial strain. For G–B–O_2_ (see Figure 5m–r), both the spin-up and spin-down states display the metal character except for the tensile strain of 6% and 10%. At a 6% strain (see Figure 5q), the spin-up and spin-down states show the semiconductor and metal character, respectively. At 10% strain (see Figure 5r), both the spin up and spin down states show a semiconductor character. For G–D–O_2_ (see Figure 5s–x), with the increasing compressive strain (see Figure 5s–v), both the spin-up and spin-down states show an indirect semiconductor character. And at a −8% compressive strain (see Figure 5s), the spin-down state turns to the direct semiconductor character. When the compressive strain is less than 2% or there is no strain (see Figure 5u–v), the spin-up and spin-down states reveal indirect and metal characters, respectively. Under the tensile strain (see Figure 5w,x), we can see that the spin-up and spin-down states reveal an indirect semiconductor character. Under the equibiaxial strain, the structure is asymmetry, which changes the orbital contributions near the Fermi. In this way, there are different strain effects, inducing the direct–indirect gap transitions. Above all, there are direct–indirect gap transitions only in G–O_2_ and G–D–O_2_. However, for the doped graphene systems (G–N–O_2_ and G–B–O_2_), they generally keep a metal character. Thus, it is a very efficient and reversible approach for regulating electronic properties of different adsorption systems by applying strain. Such interesting phenomena in the electronic structures could pave the way for elastic electronic devices working under strain and mechanical sensors on the basis of new 2D crystals. The gap opens mainly because the sublattice symmetry is broken. The honeycomb sublattices (which are shifted by a translation vector [28,29,30,31]) sustain a displacement under strain. Based on the tight-binding method [32], the structure deformation would lead to the change of the corresponding transfer integral. Accordingly, there is the Π-like band splitting, which could open the gap, making the adsorption system turns into semiconductor material [31,33,34]. As a result, this indicates that the electronic transport properties of graphene adsorption systems can be effectively regulated by external forces. The gap opening is a vital and really popular research issue for the application of 2D materials in the photoelectronics fields, where needs a proper on/off ratio. The adsorption system has to be transferred to semiconductor so that it can be used for transistors in nanoelectronics, despite its high velocity massless Fermions as charge carrier. In this way, the gap in 2D materials with a Dirac cone could be opened by the degenerate perturbation owing to the symmetry breaking.

## 4. Conclusions

In summary, we studied the equibiaxial strain effects on the interactions between the O_2_ molecule and various graphenes (G–O_2_, G–N–O_2_, G–B–O_2_, and G–D–O_2_) through applying first-principles calculations. It was found that under the same strain, G–D–O_2_ possesses the highest stability in contrast to G–O_2_, G–N/B–O_2_, and G–D–O_2_. The stability of O_2_ adsorption on various graphenes can be enhanced enormously by the compressive strain larger than 2%. However, the stability of O_2_ adsorption is reduced under the compressive strain smaller than 2% and the tensile strain. Moreover, the types of graphenes have little influence on the stress and Young’s modulus. For G–O_2_ and G–D–O_2_, the band gap direct–indirect transition occurs under equibiaxial strain. Our findings provide new approaches for designing and synthesizing elastic electronic devices based on graphene oxide. Significantly, these results do not account for the isotopic exchange studies where the isotopes of oxygen were reacted with graphene, which showed that the formation of different oxygen structures on graphene was the result of a number of sequential reactions [35].

## Figures and Tables

**Figure 1 materials-13-04945-f001:**
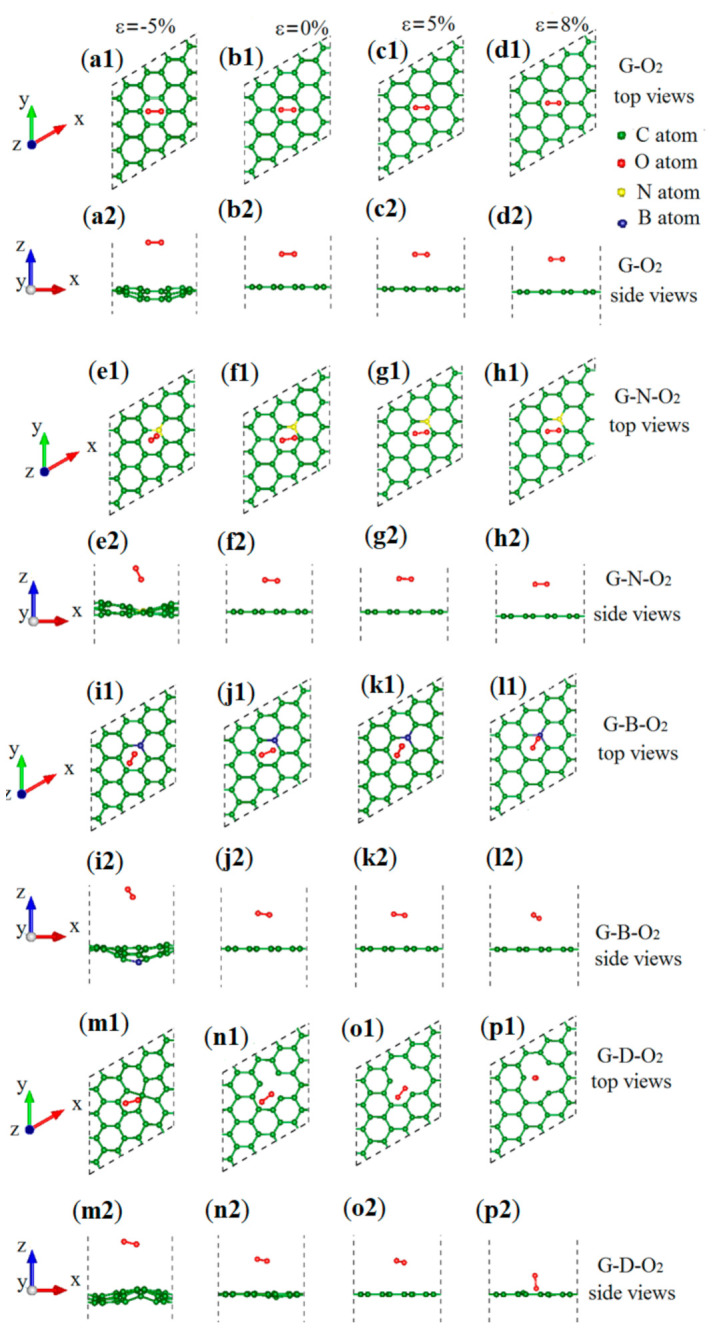
(**a1**–**d2**) Top and side views of G–O_2_; (**e1**–**h2**) G–N–O_2_; (**i1**–**l2**) G–B–O_2_; and (**m1**–**p2**) G–D–O_2_ under equibiaxial strains of −5%, 0%, 5%, and 8%. The green, red, yellow and blue balls refer to the C, O, N, and B atoms, respectively, and the insides of dashed lines denote the unit cells of the systems.

**Figure 2 materials-13-04945-f002:**
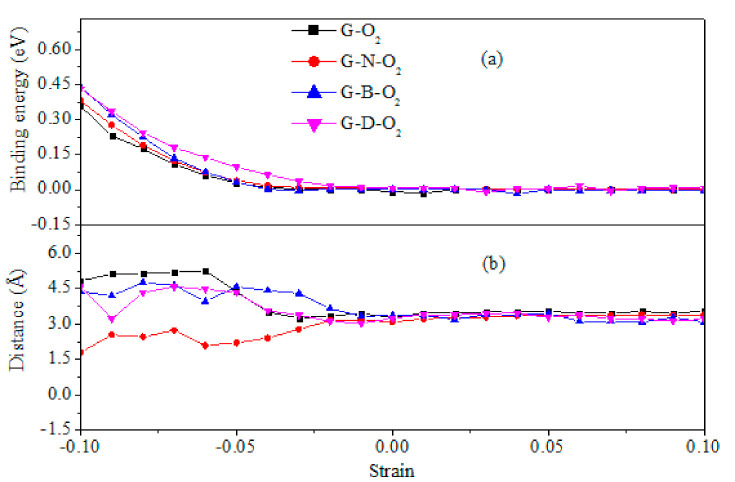
(**a**) Binding energy (E_b_) per atom as a function of equibiaxial strain for G–O_2_, G–N–O_2_, G–B–O_2_, and G–D–O_2_; (**b**) The shortest distance of the O atom to different graphenes with equibiaxial strain..

**Figure 3 materials-13-04945-f003:**
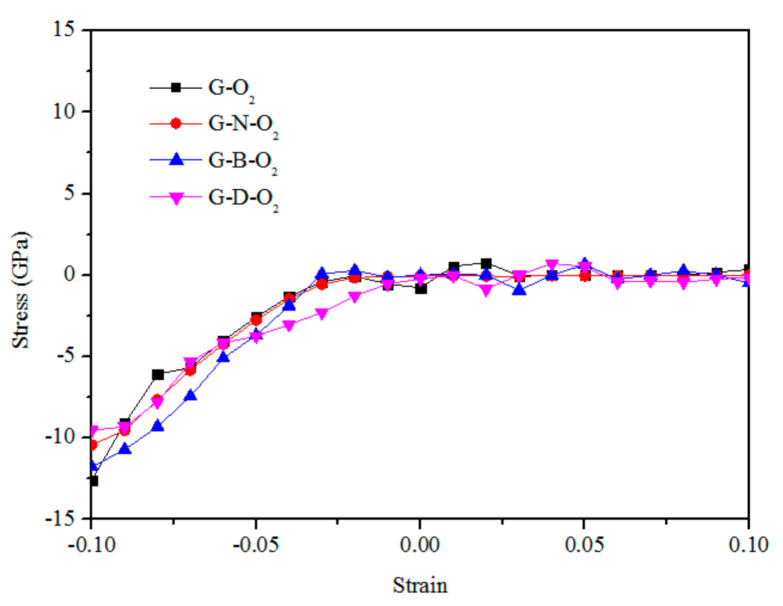
Stress as a function of equibiaxial strain for G–O_2_, G–N–O_2_, G–B–O_2_, and G–D–O_2_.

**Figure 4 materials-13-04945-f004:**
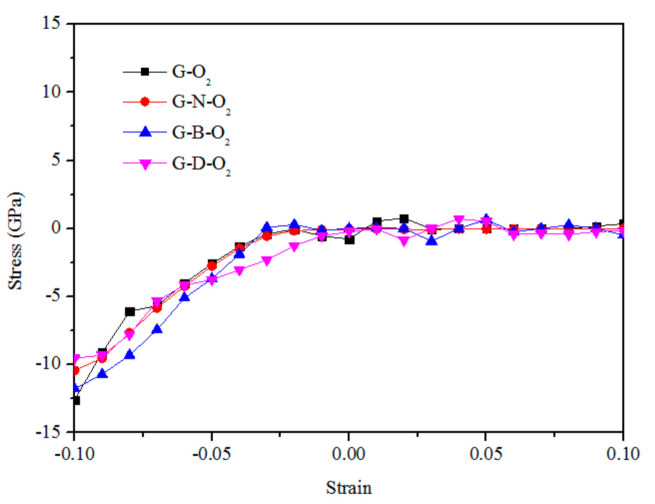
Young’s modulus as a function of equibiaxial strain for G–O_2_, G–N–O_2_, G–B–O_2_, and G–D–O_2_.

**Figure 5 materials-13-04945-f005:**
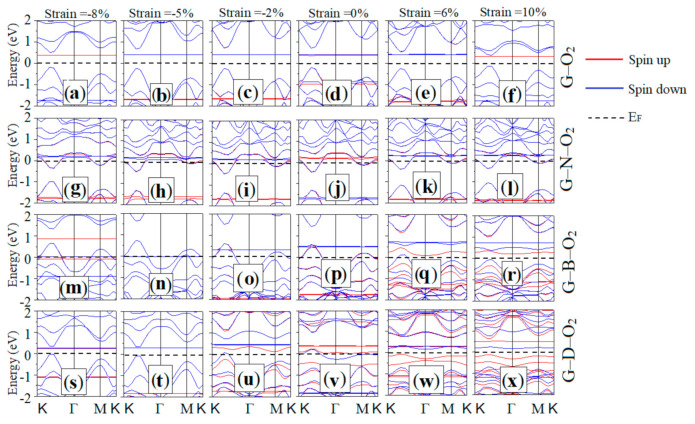
(**a**–**f**) Band structures for G–O_2_; (**g–l**) G–N–O_2_; (**m–r**) G–B–O_2_; and (**s–x**) G–D–O_2_ under equibiaxial strains. The figures from left to right columns correspond to the band structures calculated at the representative strains of −8%, −5%, −2%, 0%, 6%, and 10% in turn. The red and blue lines represent spin-up and spin-down bands, respectively. The Fermi level is set to zero and indicated by the horizontal dashed line.

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
