# Peer review of "Equibiaxial Strained Oxygen Adsorption on Pristine Graphene, Nitrogen/Boron Doped Graphene, and Defected Graphene"

_materials, 2020, doi:10.3390/ma13214945_

Round 1

Reviewer 1 Report

The manuscript regards the adsorption of oxygen molecules in different type of graphenes by means of computational methods. The VASP, shortly, Vienna Ab Initio Simulation Package is first principle method used to model large systems, like clusters or in this case graphene, HOPG etc. It allows the use of higher order perturbation theory, even up to MP2.  The authors have gathered experimental data based on strain engineering from the computational point of view, which may aid experimental researchers in potential approaches towards fabrication.    I see no potential clues pointing towards rejection of this publication, as it is in most cases well written (with few flaws, like incomplete sentence in the line 19), which can be simply corrected upon resubmission.    The interesting highlight of the manuscript is the compilation of the electronic properties, which may be of aid for any ARPES (angle resolved X Ray photoelectron spectroscopy) researchers.   To summarize: other than grammar check there are no other points of constructive critique. As the authors provided contact information, many question which may arise from this manuscript can be directly sent to the authors.

The authors have used computational methods to elucidate the peculiarities of oxygen molecule with various types of graphene by using VASP package. The results are clearly presented and of interest to the potential reader, nevertheless the figures, especially the massive figure 5 could receive a final touch in improving the image quality. From scientific points of view the authors did not mention possibility of isotopes of oxygen. Was is taken into consideration? It would enhance interest from the experimental scientists.

Reviewer 2 Report

Dear Authors

I recommend a major revision to be performed.

The research addresses the changes of structure and electronic properties of graphene-O2 complexes at different levels of stress. The work is interesting but the topic is not very original as stressed graphene has long been studied. The publication adds this oxygen complex as a novelty. In fact the publication does not open a new topic but is one of the many many works dealing with stressed/altered graphene. The paper is not well written, whole sentences do not make any sense and must be rewritten or the work to be rejected...
The conclusions are consistent with the evidence and arguments presented, but they do not represent any significant novelty. From many works it is known that corrugation and stress can cause bandgap opening.

English language must be revised. For example in the Introduction "Our work would be an in controlling the graphene-based structure"; that does not make sense.

The illustrations are very low quality, barely readable, they must be improved.

Please provide clear illustrations of the band gap opening (band gaps, etc). Explain the effect of band gap opening as precisely as possible.

Round 2

Reviewer 2 Report

Dear Authors

The manuscript has been reviewed and now it is much easier to read and understand. I can advice the manuscript to be accepted in its present form.

Only thing - if possible the quality of the figures should be improved. Especially Fig 1,3,4,5. Please improve the quality of figures if you can.

Best.